# Effects of Creatine Supplementation during Resistance Training Sessions in Physically Active Young Adults

**DOI:** 10.3390/nu12061880

**Published:** 2020-06-24

**Authors:** Scotty Mills, Darren G. Candow, Scott C. Forbes, J. Patrick Neary, Michael J. Ormsbee, Jose Antonio

**Affiliations:** 1Faculty of Kinesiology and Health Studies, University of Regina, Regina, SK S4S0A2, Canada; scottymills4@gmail.com (S.M.); Patrick.Neary@uregina.ca (J.P.N.); 2Department of Physical Education, Faculty of Education, Brandon University, Brandon, MB R7A6A9, Canada; forbess@brandonu.ca; 3Institute of Sports Sciences & Medicine, Department of Nutrition, Food, & Exercise Sciences, Florida State University, Tallahassee, FL 32313, USA; mormsbee@fsu.edu; 4Discipline of Biokinetics, Exercise and Leisure Sciences, University of KwaZulu-Natal, 4041 Durban, South Africa; 5Department of Health and Human Performance, Nova Southeastern University, Davie, FL 33314, USA; exphys@aol.com

**Keywords:** intra-workout, muscle mass, strength, endurance, power

## Abstract

The purpose was to examine the effects of creatine supplementation during resistance training sessions on skeletal muscle mass and exercise performance in physically active young adults. Twenty-two participants were randomized to supplement with creatine (CR: *n* = 13, 26 ± 4 yrs; 0.0055 g·kg^−1^ post training set) or placebo (PLA: *n* = 9, 26 ± 5 yrs; 0.0055 g·kg^−1^ post training set) during six weeks of resistance training (18 sets per training session; five days per week). Prior to and following training and supplementation, measurements were made for muscle thickness (elbow and knee flexors/extensors, ankle plantarflexors), power (vertical jump and medicine ball throw), strength (leg press and chest press one-repetition maximum (1-RM)) and muscular endurance (one set of repetitions to volitional fatigue using 50% baseline 1-RM for leg press and chest press). The creatine group experienced a significant increase (*p* < 0.05) in leg press, chest press and total body strength and leg press endurance with no significant changes in the PLA group. Both groups improved total body endurance over time (*p* < 0.05), with greater gains observed in the creatine group. In conclusion, creatine ingestion during resistance training sessions is a viable strategy for improving muscle strength and some indices of muscle endurance in physically active young adults.

## 1. Introduction

Creatine is an organic compound naturally produced in the body from reactions involving the amino acids arginine, glycine and methionine in the kidneys and liver or consumed in the diet primarily from red meat, poultry, seafood [1] or supplementation practices. There is substantial evidence that creatine supplementation and resistance training increases muscle mass and performance (i.e., strength) more than placebo and resistance training, possibly by influencing phosphate metabolism, cellular hydration status, calcium and protein kinetics, glycogen content, satellite cells, growth factors, inflammation and oxidative stress (for reviews see [2,3,4,5,6]).

Harris et al. [7] showed that prior muscle contractions augmented total intramuscular creatine uptake from creatine supplementation, possibly through an upregulation in creatine transport kinetics, an increase in sodium–potassium pump function during exercise, or by an increase in blood flow delivery of creatine to exercising muscles (for review see [8]). Ingesting creatine immediately following each set of resistance training may therefore increase creatine uptake into skeletal muscle, which over time could lead to greater gains in muscle mass and performance compared to placebo. Therefore, the purpose of this study was to examine the effects of creatine ingestion during resistance training sessions in physically active young adults. It was hypothesized that the repeated ingestion of creatine following each set of resistance training would lead to greater gains in muscle mass and performance compared to placebo immediately following each set of resistance training.

## 2. Materials and Methods

### 2.1. Participants

Physically active males and females (19–35 years of age) who had been performing structured resistance training (>3×/week for ≥6 weeks) prior to the start of the study were recruited. An a priori power analysis (G*Power v. 3.1.5.1) indicated that 34 participants were required. This calculation was based on a moderate effect size (Cohen’s d = 0.25), an alpha level of 0.05, a β-value of 0.8 for a repeated measures within or between an analysis of variance (ANOVA) design with two groups and a correlation among repeated measures value of 0.5 [9]. Participants were excluded from the study if they were taking medications that could affect muscle biology (i.e., corticosteroids), had ingested creatine monohydrate or dietary supplements containing creatine ≤4 weeks prior to the start of resistance training and supplementation, if they were vegetarian or if they had pre-existing kidney or liver abnormalities.

Participants were instructed not to change their habitual diet or engage in additional physical activity that was not part of their normal daily routine or consume non-steroidal anti-inflammatory drugs during resistance training and supplementation, as these interventions can affect muscle protein turnover [10]. The study was approved by the Research Ethics Board at the University of Regina. Participants were informed of the risks, potential benefits and purposes of the study before written consent was obtained.

### 2.2. Research Design

The study used a double-blind, placebo-controlled, repeated measures design. In order to minimize group differences, participants were matched according to age, sex and body mass. After exclusion criteria were applied, participants were randomized on a 1:1 basis to supplement with creatine monohydrate (CR) or placebo (PLA). Prior to testing, participants were instructed to refrain from alcohol and intense physical activity for 24 h and food and drink for 3 h (water was permitted ad libitum). The primary dependent variables assessed prior to and following training and supplementation were (1) muscle thickness (elbow and knee flexors/extensors, ankle plantarflexors; ultrasonography), (2) power (vertical jump and medicine ball throw), (3) strength (1-RM leg press and chest press), and (4) endurance (maximum number of repetitions performed for one set using 50% of baseline 1-RM for leg press and chest press). In addition, participants filled out a three-day food diary during the first and final week of training and supplementation to determine whether total energy (kcal) and macronutrient intake changed over time.

### 2.3. Supplementation

Creatine (Creapure^®^ AlzChem, Trostberg GmbH, Germany) and placebo (Globe^®^ Plus 10 DE Maltodextrin, Univar Canada) were in powder form. Both products were similar in taste, color (white), texture and appearance. The purity of Creapure^®^ was established at >99.9% by independent laboratory testing (The Cary Company, Addison, IL., USA). Two individuals not involved in any other aspect of the study were responsible for randomizing participants into groups and for preparing participant study kits which included their supplement for the duration of the study, detailed supplementation instructions, measuring spoons and a water bottle. The creatine supplementation dosage was 0.1 g·kg^−1^·d^−1^ as this dosage has been shown to be effective, when combined with resistance training, for increasing muscle mass and muscle performance [11]. On training days (five days/week), creatine and placebo were mixed with water (900 mL) and participants consumed 50 mL of the solution containing 0.0055 g·kg^−1^ of creatine or placebo immediately after each set of resistance training (18 sets per training day). Creatine and placebo were consumed in a plastic shaker bottle with gradations (mL) on the side to ensure that 50 mL of the solution was consumed after each set. Participants were instructed to refrain from food or drink (water was permitted ad libitum) for 1 h before and after each training session so that a valid estimate of the effects of intra-workout creatine supplementation could be made. On the non-training days (two days per week), participants refrained from consuming creatine or placebo as the purpose of the study was to investigate the effects of creatine ingestion during resistance training sessions. Adherence with the supplementation protocol was assessed by a compliance log. Upon completion of the study, participants were asked whether they thought they were administered creatine, placebo, or unsure about what supplement they consumed.

### 2.4. Resistance Training Program

Participants followed the same periodized resistance training program for six weeks. Resistance training started on the first day of supplementation and consisted of a four-day split routine involving three sets to volitional fatigue per exercise (Set 1: ~6 repetitions, Set 2: ~8 repetitions, Set 3: ~10 repetitions), with 2 min rest between sets. The load was adjusted (if necessary) following each set to achieve volitional fatigue at the desired number of repetitions. We have used a similar training program successfully to increase muscle mass and performance in physically active young adults [12]. Day 1 involved leg and core musculature and included the following exercises in order: barbell back squat, walking dumbbell lunge, leg extension, leg curl, calf raise and weight abdominal crunch. Day 2 involved upper-body and core and included the following exercises in order: flat barbell bench press, flat bench dumbbell chest fly, overhead cable triceps extension with rope, dumbbell curl, dumbbell concentration curls and pallof press. Day 3 was a rest day from training. Day 4 involved leg and core musculature and included the following exercises in order: leg press, dumbbell goblet squat, dumbbell reverse lunge, leg curl, calf raise, and cable crunches. Day 5 involved upper-body musculature and included the following exercises in order: flat barbell chest press, barbell row, dumbbell shoulder press, dumbbell skull crushers, dumbbell hammer curls, and triceps extensions with rope. Day 6 served as a rest day from training. This cycle was repeated for six weeks. Participants filled out resistance training logs after each training session to determine adherence and compliance to the study and to determine total training volume (sets × repetitions × load; kg) performed over time.

### 2.5. Muscle Thickness

Muscle thickness (right side) of the elbow and knee flexors and extensors and ankle plantarflexors was measured using B-mode ultrasound (LOGIQ e, GE Medical Systems, China) as previously described [13]. To help ensure that exercise induced muscle swelling (edema) did not influence the results, participants were instructed not to perform resistance training for 48 h prior to baseline measurements. Post-testing measurements were obtained 48 h after the last training session of the study. The reproducibility of muscle thickness measurements was determined by assessing eight participants on two separate days (24 h apart). The coefficients of variation (CV) and intraclass correlation coefficients (ICC) were: elbow flexors (CV: 4.4%, ICC: 0.993), elbow flexors (CV: 7.1%, ICC: 0.878), knee flexors (CV: 5.4%, ICC: 0.936), knee extensors (CV: 2.9%, ICC: 0.991), and ankle plantarflexors (CV: 2.8%, ICC: 0.976). The same researcher performed all measurements.

### 2.6. Muscle Performance

Power, strength, and endurance were assessed in the following order: (a) vertical jump, (b) medicine ball throw, (c) leg press 1-RM, (d) chest press 1-RM, (e) leg press endurance, and (f) chest press endurance. Each test was separated by 5 min of rest. Prior to the start of testing, participants performed a 5 min warm-up on a stationary cycle ergometer (Ergomedic 828 E, Monark, GIH Sweden) at a self-selected intensity and completed light stretching. The same researcher demonstrated how to properly perform each test. To determine lower-body power, a vertical jump test was used. Participants standing reach height was measured followed by three vertical jump tests to displace the Vertec vanes. Rest time between jump tests was 30 s. Participants peak power was calculated from the highest of the three vertical jump trials using Sayers Peak power equation [14] (Peak power [Watts; W] = [51.9 × VJ (vertical jump) height (cm)] + [48.9 × Body mass (kg)] − 2007). To measure upper-body power, a medicine ball throw test was used. Participants stood behind a line marked on the floor in a standing position and were instructed to throw the medicine ball (13.6 kg) using both hands with fingers pointed in from chest level, similar to a chest pass in basketball, as far as they could horizontally. Participants were further instructed not to use their lower body for power generation and to not step over the line after the medicine ball was released. Participants performed three trials, with their longest throw being used for analysis. Rest time between trials was 30 s.

Detailed procedures for determining leg press and chest press 1-RM strength are previously described [12]. These two exercises were chosen as a measurement of strength because they involve the major muscle groups in the lower and upper body [15]. The CV’s and ICC’s were 3.8% and 0.99 for leg press 1-RM and 3.1% and 0.99 for chest press 1-RM [16]. To determine leg press and chest press endurance, participants performed one set of repetitions to volitional fatigue (defined as the inability to perform the concentric phase of a muscle contraction) using 50% baseline 1-RM for the leg press and chest press.

### 2.7. Diet

Average total energy (kcal) and macronutrient intake for three days (two weekdays and one weekend day) was determined during the first and final week of supplementation and resistance training. Participants used a three-day food booklet to record all food items, including portion sizes consumed, for the three designated days. MyFitnessPal, which shows good validity compared to paper-based food records [17], was used to analyze three-day food records.

### 2.8. Adverse Events Assessment

In the case of an adverse event, participants were required to complete an adverse event form in order to provide details on the type of adverse event, the severity (i.e., mild, moderate, severe, or life threatening), the frequency, and the relationship to the intervention (i.e., not related, unlikely, possible, probable, or definite).

### 2.9. Statistical Analyses

The primary analysis performed was a 2 (groups: creatine vs. placebo) × 2 (time: pre-training vs. post-training) repeated measures ANOVA to determine differences between groups over time for changes in muscle thickness, power, strength, endurance, and diet. As a secondary analysis, a 2 (groups: creatine vs. placebo) × 2 (sex: males vs. females) × 2 (time: pre-training vs. post-training) repeated measures ANOVA was performed on all variables to determine differences between males and females. If significant interactions were detected using ANOVA testing, file splitting and paired sample t-tests were performed to determine where differences occurred between means. A one-factor ANOVA was used to assess baseline data, training volume, and absolute change scores. Significance was set a priori at an alpha level of *p* < 0.05. Cohen’s *d* effect size (ES) was calculated as post-training mean minus pre-training mean/pooled pre-training standard deviation mean [18]. An ES of 0.00–0.19 was considered trivial, 0.20–0.49 was considered small, 0.50–0.79 was considered moderate, and ≥0.80 was considered large. Statistical analyses were performed using IBM^®^ SPSS^®^ Statistics, v. 26 (Chicago, IL, USA).

## 3. Results

### 3.1. Participants

Based on the sample size calculation, our recruitment goal was 34 participants. However, we were only able to enroll and randomize 26 eligible participants (13 male, 13 female) into the study, which started in August 2019 and ended in December 2019, because the vast majority of participants could only commit to the study during this time frame (see Figure 1 for a summary of recruitment, allocation and analysis). Following randomization, four females (all from the PLA group) withdrew due to time constraints (*n* = 2), doctor recommendations (*n* = 1) and personal injury (*n* = 1), all unrelated to the study. Therefore, 22 participants (CR = 13, (7 male, 6 female); PLA = 9 (6 male, 3 female)) completed the study. One female in the creatine group reported gastrointestinal irritation during the first week of creatine supplementation but this did not result in her withdrawing from the study. Seventeen participants (CR = 9, (6 male, 3 female); PLA = 8 (5 male, 3 female)) were able to provide three-day food records (first and final week of training and supplementation).

Following the intervention, participants were asked whether they thought they were administered creatine, placebo, or unsure about what supplement they consumed. In the CR group, 10 participants correctly guessed they were consuming creatine and three did not know. In the PLA group, five participants correctly guessed they were consuming placebo and four did not know. Training compliance (CR: 25/28 sessions completed or 90.9%; PLA: 25/28 sessions completed or 89.3%) and supplementation compliance (CR: 27/28 sessions or 97.58%; PLA: 28/28 sessions or 100%) were similar between groups over time (*p* > 0.05).

Baseline data are presented in Table 1. Both groups experienced similar increases in body mass over time (CR: pre 80.55 ± 18.07 kg, post 81.72 ± 17.44 kg; PLA: pre 79.88 ± 19.97 kg, post 80.14 ± 18.95 kg; *p* = 0.05; observed power = 0.51). There was a group x time interaction (*p* = 0.016) for total energy intake (*d* = 0.72; observed power = 0.72; Table 2). The creatine group significantly decreased total energy intake over time with no change in the placebo group. There were no changes over time for carbohydrate, fat, protein or relative protein intake. There were no significant differences between groups for total training volume performed over time (CR: 315,376 kg [95% CI: 222,936, 407,816]; PLA: 407,549 kg [95% CI: 326,615, 488,483]; *p* = 0.134). There was a sex main effect (*p* < 0.05) for body mass and all measures of muscle thickness (except ankle plantarflexors, *p* = 0.056), strength, and power, with males having higher values compared to females. There were no differences between males and females for measures of muscle endurance (*p* > 0.05).

### 3.2. Muscle Thickness

There was a time main effect (*p* < 0.05) for the elbow flexors, elbow extensors, knee extensors, knee flexors, and all muscle groups combined (Table 3), with no significant differences between groups. There was no change over time for the ankle plantarflexors (*p* = 0.471).

### 3.3. Muscle Performance

There was a group × time interaction for leg press (*p* = 0.025, *d* = 0.33, observed power = 0.63; Figure 2A), chest press (*p* = 0.012, *d* = 0.36, observed power = 0.75; Figure 2B) and total body strength (leg press and chest press combined: *p* = 0.03; *d* = 0.18, observed power = 0.89; Figure 2C). Post hoc analyses showed that the creatine group experienced a significant increase in strength over time, with no significant changes in the placebo group.

There was a group × sex x time interaction (*p* = 0.039) for chest press strength. Males on creatine experienced a significant increase over time (CR: pre 166.53 ± 22.82 kg, post 187.59 ± 18.64 kg; *d* = 0.47, observed power = 0.55) with no change for males on placebo (pre 181.05 ± 38.71 kg, post 180.30 ± 43.07 kg). There was no significant change over time for females on creatine (pre 67.66 ± 16.87 kg, post 72.19 ± 14.65 kg, *p* = 0.203) or placebo (pre 68.03 ± 20.78 kg, post 69.55 ± 23.27 kg, *p* = 0.203).

There was a group × time interaction for leg press (*p* = 0.013, *d* = 0.95, observed power = 0.74; Figure 3A) and total body endurance (leg press and chest press combined: *p* = 0.04, *d* = 0.96, observed power = 0.87; Figure 3C). Post hoc analyses indicated that the creatine group significantly increased the number of repetitions performed over time for the leg press with no significant change in the placebo group (Figure 3A). Both groups increased total body endurance over time, but the improvement was greater in the creatine group (Figure 3C). Both groups experienced a similar change (*p* < 0.05) in the number of repetitions performed over time for the chest press (Figure 3B).

Regarding muscle power, there was a significant time main effect (*p* < 0.05) for vertical jump (CR: pre 4387.08 ± 1126.50 W, post 4629.80 ± 1161.66 W, PLA: pre 4561.87 ± 1367.64 W, post 4813.28 ± 1390.57 W), medicine ball throw (CR: pre 386.15 ± 103.58 cm, post 408.94 ± 109.19 cm; PLA: pre 386.76 ± 103.12 cm, post 396.51 ± 96.90 cm) and total body power (vertical jump and medicine ball throw combined) (CR: pre 4539.12 ± 1161.86, post 4790.80 ± 1198.50; PLA: pre 4714.15 ± 1404.32, post 4969.39 ± 1421.15; *p* < 0.05), with no significant differences between groups.

## 4. Discussion

This was the first study to examine the effects of creatine supplementation during resistance training sessions on muscle accretion and performance. Results showed that creatine ingestion only on training days produced greater gains in muscle strength and endurance (except chest press) compared to placebo in a very small cohort of physically active young adults (Figure 2 and Figure 3). Males on creatine improved chest press strength over time with no change for females on creatine. One participant reported gastrointestinal irritation during the first week of creatine supplementation. No other participant reported an adverse event. Overall, creatine ingestion during resistance training sessions is a well-tolerated, viable strategy for improving muscle strength and endurance.

The significant increase in muscle strength (leg press: Δ 43 ± 32 kg, chest press Δ 13 ± 11 kg; Figure 2) from creatine supplementation during resistance training sessions is comparable to our previous studies showing significant improvements in muscle strength from creatine supplementation immediately before and immediately after resistance training sessions in young (creatine before: chest press Δ 7 ± 8 kg; creatine after: chest press Δ 8 ± 6 kg; [19]) and older adults (creatine before: leg press Δ 36 ± 26 kg, chest press Δ 15 ± 13 kg; creatine after: leg press Δ 40 ± 38 kg, chest press Δ 15 ± 12 kg; [11]). Collectively, results across studies indicate that creatine supplementation prior to, during and following resistance training sessions are effective ingestion strategies to improve muscle strength. However, it remains unknown which pattern of creatine ingestion (pre-exercise vs. during exercise vs. post-exercise) would produce the greatest muscle benefits in young and older adults.

The greater increase in muscle strength (Figure 2) and endurance (Figure 3) from creatine supplementation supports the findings of several meta-analyses and review articles [4,20,21,22,23]. While the mechanistic actions of creatine were not measured in this study, creatine supplementation has been shown to increase intramuscular PCr levels which may have accelerated ATP resynthesis and/or PCr recovery following each set. Over time, this may have contributed to the greater gains in strength and endurance. It is also possible that creatine supplementation augmented calcium reuptake into the sarcoplasmic reticulum which would result in faster actin-myosin cross-bridge cycling during repeated muscle contractions [24] leading to improvements in muscle strength and endurance over time. Furthermore, creatine may have influenced muscle glycogen stores. Glycogen increases ATP resynthesis during resistance training sessions [4] and, unfortunately, glycogen depletion occurs with resistance training [25]. Creatine supplementation has been shown to increase the translocation and content of GLUT-4 transport proteins in adults performing resistance training compared to placebo [26]. Potentially, an increase in GLUT-4 content would increase glucose disposal and attenuate glycogen depletion during training sessions. It is somewhat puzzling that muscle strength and leg press endurance did not increase over six weeks of training in the placebo group. However, with only nine participants, we likely had insufficient power to detect significant changes over time.

The greater increase in chest press strength in males compared to females on creatine may be related to initial PCr levels and muscle protein catabolism. Some research suggests that females may have higher initial resting creatine levels than males [27]. A main variable which dictates an individual’s responsiveness to creatine supplementation in initial levels of intramuscular creatine [2]. In addition, females do not appear to experience a reduction in whole-body or muscle protein breakdown from creatine supplementation [28,29]. These potential sex differences in creatine metabolism may have influenced chest press results over time. However, no measure of intramuscular creatine or muscle protein catabolism was performed so we can only speculate regarding the sex difference in chest press strength. It is important to note that females on creatine experienced a ~5 kg increase in chest press strength over time. Our small sample size may have decreased our ability to detect significant differences with training.

Creatine supplementation had no greater effect on muscle accretion (Table 3) compared to placebo. The lack of significant findings may be related to the short duration of training and supplementation (six weeks), intermittent creatine dosing protocol, decrease in total energy intake over time and low sample size. In the largest and most comprehensive meta-analysis performed to date, creatine supplementation resulted in greater gains in muscle accretion when the study period was ≥10 weeks in duration [4]. Furthermore, due to the objectives of the study, participants only ingested creatine on training days (five days/week). Perhaps daily creatine ingestion during a resistance training program is required to produce statistically significant greater gains in muscle accretion compared to placebo. The reduction in total energy intake in the creatine group may have also masked any effects of creatine on muscle accretion. Finally, our small sample size likely reduced our ability to detect significant differences between groups for muscle thickness. Future research should investigate the mechanistic actions of daily creatine supplementation (includes intra-workout and non-training days) and a longer training period (>6 weeks) on muscle accretion in physically active young trained adults.

In addition to our small sample size, there were other limitations to the study which may have influenced our findings. Three-day food records do not measure habitual dietary intake of creatine. A high intake of red meat, seafood or poultry may have influenced the responsiveness to creatine supplementation. Furthermore, food records typically have high variability due to the participant’s memory and accuracy in recording and reporting correct portion sizes and frequency of food intake [30]. Finally, there was no measure of neuromuscular activation, muscle fiber morphology or recruitment, muscle protein kinetics, satellite cells, growth factors, hormones, oxidative stress or inflammation.

## 5. Conclusions

Creatine ingestion during resistance training sessions is a safe and effective strategy to increase muscle strength and endurance in physically active young adults. However, it is unknown whether intra-workout creatine supplementation is more beneficial than consuming creatine at other times of the day during a resistance training program. Future research should determine the effects of the timing of creatine supplementation on muscle mass and performance in a large cohort of physically active young adults.

## Figures and Tables

**Figure 1 nutrients-12-01880-f001:**
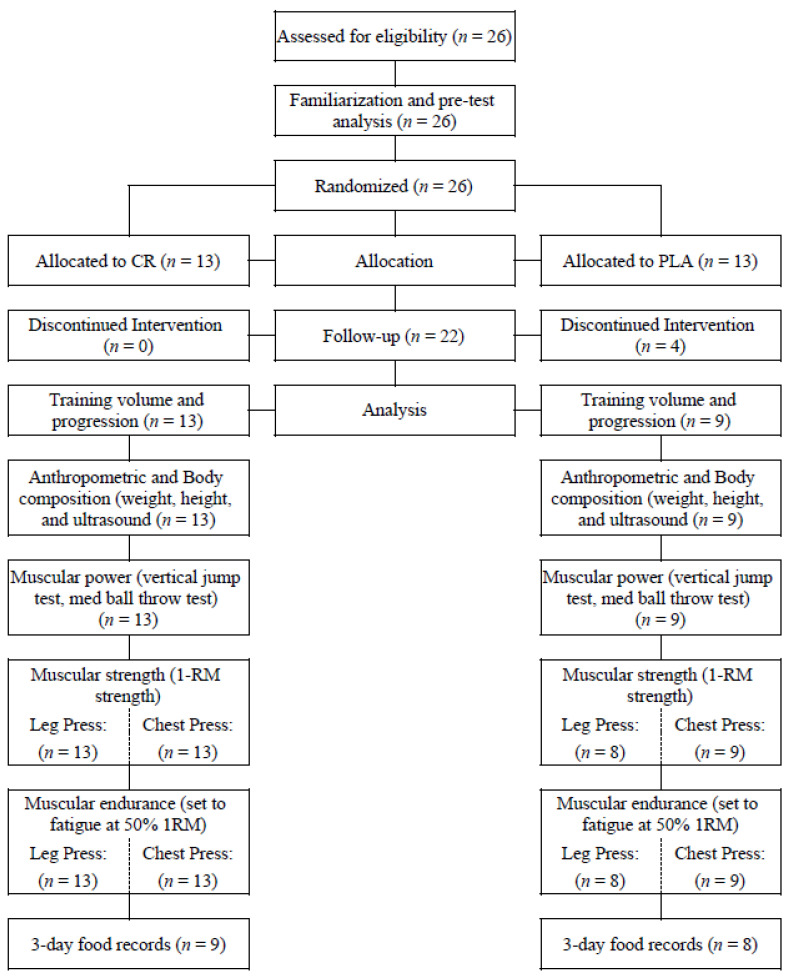
Summary of recruitment, allocation and analyses.

**Figure 2 nutrients-12-01880-f002:**
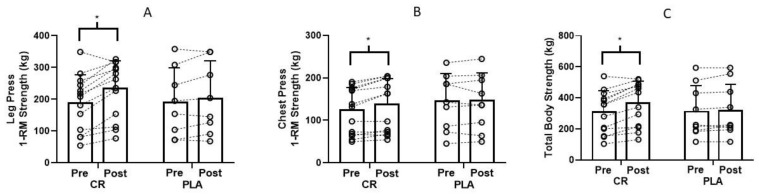
(**A**) Leg press (**B**) Chest press (**C**) Total body strength (pre and post training) for CR (*n* = 13) and PLA (*n* = 8) groups. Each dot represents an individual participant. There was a significant group x time interaction. Post hoc analyses showed that the CR group increased over time with no changes in the PLA group. * Significantly different than baseline.

**Figure 3 nutrients-12-01880-f003:**
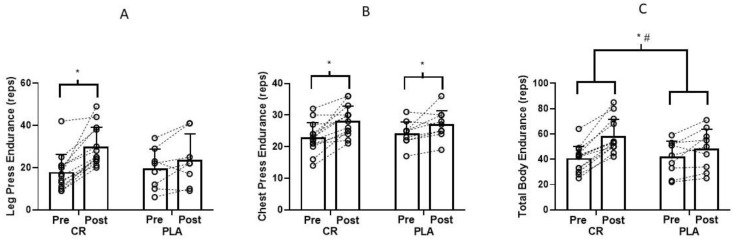
(**A**) Leg press (**B**) Chest press (**C**) Total body endurance (leg press and chest press combined; pre and post training) for CR (*n* = 13) and PLA (*n* = 8) groups. Each dot represents an individual participant. There was a significant group × time interaction for (**A**,**C**) and a significant main effect of time for (**B**). (**A**) Post hoc analyses showed that the CR group increased over time with no changes in the PLA group. (**B**) Both groups increased over time with no differences between groups. (**C**) Both groups increase over time, however the increase was greater in the CR group compared to PLA. * Significant change over time; # CR significantly greater than PLA.

**Table 1 nutrients-12-01880-t001:** Baseline characteristics.

	Creatine (*n* = 13)	Placebo (*n* = 9)	*p*-Value
Age (yrs)	26.15 (4.66)	26.44 (5.10)	0.891
Mass (kg)	80.5 (18.07)	79.88 (19.97)	0.936
Height (cm)	174.50 (10.53)	175.05 (12.09)	0.911
Muscle thickness (cm)			
Elbow flexors	2.97 (0.88)	3.03 (0.93)	0.87
Elbow extensors	2.76 (0.78)	2.84 (0.65)	0.805
Knee extensors	3.92 (0.93)	3.82 (0.45)	0.783
Knee flexors	3.36 (0.57)	3.34 (0.53)	0.959
Ankle plantarflexors	3.53 (0.41)	3.12 (0.50)	0.051
Total muscle thickness	16.55 (2.50)	16.18 (2.28)	0.726
Muscle strength (kg)			
Chest press	120.89 (54.8)	143.38 (65.10)	0.391
Leg press	188.06 (88.68)	188.06 (107.96)	0.987
Total strength	308.96 (135.32)	326.87 (169.64)	0.792
Muscle endurance (repetitions)			
Chest press endurance	22.76 (4.96)	24.22 (3.96)	0.474
Leg press endurance	17.53 (8.96)	19.25 (9.63)	0.684
Total endurance	40.30 (10.52)	43.75 (12.04)	0.499
Muscle power			
Vertical jump (Watts)	4387.08 (1126.50)	4561.87 (1367.64)	0.746
Medicine ball throw (cm)	386.15 (103.58)	386.76 (103.12)	0.989
Total power	4539.12 (1161.86)	4714.15 (1404.32)	0.753
Diet			
Total calories (kcal/day)	2348.31 (736.06)	2148.30 (625.60)	0.558
Carbohydrate (g/day)	254.23 (94.85)	239.26 (80.61)	0.57
Fat (g/day)	93.46 (46.86)	72.45 (29.42)	0.293
Protein (g/day)	112.52 (54.28)	134.78 (60.59)	0.437
Relative protein (g/kg)	1.34 (0.68)	1.60 (0.50)	0.404

Values are means (standard deviation).

**Table 2 nutrients-12-01880-t002:** Mean absolute changes (95% confidence intervals) from baseline to six weeks for total calories (kcal/day), macronutrients (carbohydrate, fat, protein; grams/day) and relative protein (g/kg body mass).

	Creatine (*n* = 9)	Placebo (*n* = 8)	Time	Group	Interaction	*d*
			*p*-Value	*p*-Value	*p*-Value	
Total calories	−515.6 (−838.1, −193.1) *	−17.1 (−285.4, 251.1)	0.011	0.861	0.016	0.72
Carbohydrate	−39.3 (−81.5, 2.9)	−4.6 (−29.5, 20.4)	0.063	0.84	0.134	0.39
Fat	−33.4 (−69.0, 2.2)	1.7 (−21.4, 24.8)	0.113	0.786	0.082	0.84
Protein	−14.5 (−33.1, 4.2)	−3.5 (−18.3, 11.3)	0.105	0.82	0.311	0.19
Relative protein	−0.18 (−0.40, 0.03)	−0.03 (−0.22, 0.14)	0.104	0.255	0.265	0.32

* Creatine significantly different than placebo (*p* < 0.05). *d* = effect size.

**Table 3 nutrients-12-01880-t003:** Mean absolute changes (95% confidence intervals) from baseline to six weeks for muscle thickness (cm).

	Creatine (*n* = 13)	Placebo (*n* = 9)	Time	Group	Interaction	*d*
			*p*-Value	*p*-Value	*p*-Value	
Elbow flexors	0.49 (0.28, 0.70)	0.36 (−0.17, 0.90)	0.001	0.992	0.59	0.14
Elbow extensors	0.29 (0.06, 0.52)	0.21 (0.02, 0.41)	0.002	0.892	0.591	0.1
Knee extensors	0.43 (0.22, 0.65)	0.33 (0.05, 0.61)	<0.001	0.686	0.513	0.12
Knee flexors	0.27 (−0.11, 0.66)	0.25 (−0.02, 0.52)	0.034	0.908	0.919	0.05
Ankle plantarflexors	−0.05 (−0.31, 0.19)	0.18 (−0.07, 0.43)	0.471	0.101	0.168	0.11
Total body	1.44 (0.63, 2.28)	1.35 (0.41, 2.28)	<0.001	0.684	0.874	0.37

Total body = all muscle groups combined. *d* = effect size.

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
