# Peer review of "Effects of Creatine Supplementation during Resistance Training Sessions in Physically Active Young Adults"

_nutrients, 2020, doi:10.3390/nu12061880_

Round 1

Reviewer 1 Report

Based on previous studies that creatine supplementation and resistance training could increase muscle mass and performance, this study investigated whether the intra-workout creatine supplementation at repeated smaller dosages also had such an efficacy. It is interesting, but the work involved in this study seems to be too preliminary.

Major revisions:

- The quantity of patients involved in this study was too limited and not enough to draw the conclusions.

- This study only measured the changes of muscle hypertrophy and performance pre- and post-creatine supplementation, which were just the macroscopic variability. It would be more meaningful to explore the changes or mechanisms of action at cellular and molecular levels.

- Previous studies reported that a bolus dosage of creatine supplementation immediately before or after resistance training session could enhance muscle mass and strength. Authors should explain what are the advantages or disadvantages between the previous ingestion strategies and the present work strategy.

Minor revisions:

- Line 62: “alpha level of .05” should be “alpha level of 0.05”?

- Line 204: “Cr” should be “CR”.

- Line 250: extra “;” behind 43.07 kg should be deleted.

Author Response

The quantity of patients involved in this study was too limited and not enough to draw the conclusions.

Response: We agree with the reviewer and have highlighted throughout the manuscript that our small sample size was a limitation and may have influenced our results.

This study only measured the changes of muscle hypertrophy and performance pre- and post-creatine supplementation, which were just the macroscopic variability. It would be more meaningful to explore the changes or mechanisms of action at cellular and molecular levels.

Response: We agree with the reviewer. However, we did not have the ability to measure mechanistic actions of creatine in this study.

 Previous studies reported that a bolus dosage of creatine supplementation immediately before or after resistance training session could enhance muscle mass and strength. Authors should explain what are the advantages or disadvantages between the previous ingestion strategies and the present work strategy.

Response: We have revised the introduction and discussion to reflect more closely the purpose of the study which was to investigate the effects of creatine during resistance training sessions, not creatine before or after resistance training sessions.

Line 62: “alpha level of .05” should be “alpha level of 0.05”?

Response: This has been changed.

Line 204: “Cr” should be “CR”.

Response: This has been changed.

Line 250: extra “;” behind 43.07 kg should be deleted.

Response: This has been changed.

Reviewer 2 Report

In this manuscript, the authors demonstrated that intra-workout creatine ingestion is a safe and effective strategy to increase muscle strength and endurance in physically active young adults. These findings were obtained from twenty-two participants that were randomized to supplement with creatine or placebo during 6 weeks of resistant training. They showed that the creatine group experienced a significant increase in leg press, chest press and total body strength and leg press endurance compared to the placebo group. Data presented are sound and manuscript has been well written with adequate references. Although a demerit of this work is that most of data has been acquired from insufficient number of participants, the reviewer believes that this finding would be beneficial to understanding of appropriate and efficient timing of creatine ingestion during exercises. The reviewer provides following comments for improvement of the manuscript.

Comments:

  1. From my point of views, the most lacking part of this manuscript is that if you want to emphasize the effects of intra-workout creatine supplementation, you should have compared the group of intra-workout ingestion with those ingesting creatine immediately before or immediately after the training. It is recommended to analyze and compare quantitatively the effects depending upon the ingestion strategies using the published data of others in discussion section even if the authors can't do the experiment again
  2. Since the data presented in Figures 2 and 3 are mostly gained from limited number of participants, statistical analysis should be very much specific for each result. Statistical analysis described in the results section is somewhat complicated to find and employ the corresponding data. Therefore, the reviewer suggests that corresponding statistical methods employed in each figure should be described in each figure legend.
  3. In fact, the contents of discussion on results shown in Figure 2 and 3 appear to be somewhat insufficient and descriptive in the Discussion section. It would contribute to strengthen this paper if the authors could do so. In addition, the discussion is also needed about a significant decrease of total energy intake in the creatine group with no change in the placebo group.
  4. The authors described that there were no sex differences for muscle thickness, strength, and endurance or power, which is hard to believe. Therefore, you’d better show the data before and after the training delineating males and females.
  5. In Figure 1, the resolution of typeface is bit blurred. This needs to be resolved in the revised version. Or you can remake it for easy recognition. Tables 1 and 2, uniformity between upper and lower capital letters is required.
  6. Only three key words (muscle mass; power; endurance) have been provided: if this is not a regulation of this journal, at least five words are suggesting as proper key words. Why a word ‘creatine’ is not shown in the list, for example?

Author Response

From my point of views, the most lacking part of this manuscript is that if you want to emphasize the effects of intra-workout creatine supplementation, you should have compared the group of intra-workout ingestion with those ingesting creatine immediately before or immediately after the training. It is recommended to analyze and compare quantitatively the effects depending upon the ingestion strategies using the published data of others in discussion section even if the authors can't do the experiment again

Response: We have added a paragraph in the discussion comparing the strength results from the present study to our previous studies in young and older adults from creatine immediately before and after resistance training sessions.

Insert Lines 302-312: The significant increase in muscle strength (leg press: Δ 43 ± 32 kg, chest press Δ 13 ± 11 kg; Figure 2) from creatine supplementation during resistance training sessions is comparable to our previous studies showing significant improvements in muscle strength from creatine supplementation immediately before and immediately after resistance training sessions in young (creatine before: chest press Δ 7 ± 8 kg; creatine after: chest press Δ 8 ± 6 kg; [7]) and older adults (creatine before: leg press Δ 36 ± 26 kg, chest press Δ 15 ± 13 kg; creatine after: leg press Δ 40 ± 38 kg, chest press Δ 15 ± 12 kg; [9]). Collectively, results across studies indicate that creatine supplementation prior to, during and following resistance training sessions are effective ingestion strategies to improve muscle strength. However, it remains unknown which pattern of creatine ingestion (pre-exercise vs. during-exercise vs. post-exercise) would produce the greatest muscle benefits in young and older adults. 

Since the data presented in Figures 2 and 3 are mostly gained from limited number of participants, statistical analysis should be very much specific for each result. Statistical analysis described in the results section is somewhat complicated to find and employ the corresponding data. Therefore, the reviewer suggests that corresponding statistical methods employed in each figure should be described in each figure legend.

            In fact, the contents of discussion on results shown in Figure 2 and 3 appear to be somewhat insufficient and descriptive in the Discussion section. It would contribute to strengthen this paper if the authors could do so.

Response: We have inserted detailed descriptions of the statistical analyses used in each figure legend. We believe this adds clarity and understanding to the figures without reverting back to the methods and results section.

            We have added in comparison data to other studies evaluating the effect of creatine on muscle strength and have indicated which parts of the discussion are referring to Figures 2 and 3.

In Figure 1, the resolution of typeface is bit blurred. This needs to be resolved in the revised version. Or you can remake it for easy recognition.

Response: The clarity of the image has been improved.

Tables 1 and 2, uniformity between upper and lower capital letters is required

Response: Tables 1 and 2 have been carefully reviewed and adjusted for consistency.

In addition, the discussion is also needed about a significant decrease of total energy intake in the creatine group with no change in the placebo group.

Response: Insert, Lines 348-349: The reduction in total energy intake in the creatine group may have also masked any effects of creatine on muscle accretion.

The authors described that there were no sex differences for muscle thickness, strength, and endurance or power, which is hard to believe. Therefore, you’d better show the data before and after the training delineating males and females.

Response: Insert, Lines 237-241: There was a sex main effect (p < 0.05) for body mass and all measures of muscle thickness (except ankle plantarflexors, p = 0.056), strength, and power, with males having higher values compared to females. There were no differences between males and females for measures of muscle endurance (p > 0.05).

 Only three key words (muscle mass; power; endurance) have been provided: if this is not a regulation of this journal, at least five words are suggesting as proper key words. Why a word ‘creatine’ is not shown in the list, for example?

Response: We have revised the keyword section to: Intra-workout, Muscle mass; Strength, Endurance, Power. Keywords should not contain any word mentioned in the title.

Reviewer 3 Report

The aim of this study was to examine the effects of intra-workout creatine supplementation on skeletal muscle mass and exercise performance in physically active young adults. Considering the emphasis of the Introduction on creatine timing relative to training, the design of this study seems a little odd to me, particularly the use of placebo as a control. Regardless of whether creatine was provided intra-workout, or before or after, we would expect to see an increase in muscle creatine and phosphorylcreatine surely? The design of this study simply shows that creatine supplementation 5 x/week can lead to greater strength improvements over placebo. It seems the authors should have investigated whether intra-workout was superior to supplementation at other times of the day (or a standard supplementation strategy). I suggest the authors perhaps focus a little more on the true novelty here, namely their strategy of 5 days/ week supplementation on training days using micro-dosing rather than highlight the intra-workout strategy so much.

The authors report a calculated sample size requirement of 34 participants, but only 26 were recruited. Why the disparity? I think this might have limited results.

It is strange that the placebo group did not improve performance in several exercises (eg chest press, leg press). The authors state that they have previous shown this training protocol to be effective. I believe the data might be limited by the small sample size in relation to their calculated N (which would have been 17 per group).

Could the lack of an effect in females for chest press strength (lines 250-252) be due to a lack of statistical power? Certainly, a +5 kg increase seems fairly substantial.

I do think the authors should highlight a little more the potential issue of their sample size in the discussion. They make one brief mention but based upon my comments here, I think it may have substantially compromised some of the findings.

Line 208: “28/27 for supplement compliance” – how is this possible? Is this a typo?

Conclusions – I suggest also including that future work needs to elucidate the effects of timing on creatine content of muscle.

Author Response

I suggest the authors perhaps focus a little more on the true novelty here, namely their strategy of 5 days/ week supplementation on training days using micro-dosing rather than highlight the intra-workout strategy so much.

Response: Insert Lines 294-301: This was the first study to examine the effects of creatine supplementation during resistance training sessions on muscle accretion and performance. Results showed that creatine ingestion only on training days produced greater gains in muscle strength and endurance (except chest press) compared to placebo in a very small cohort of physically active young adults (Figures 2 and 3). Males on creatine improved chest press strength over time with no change for females on creatine. One participant reported gastrointestinal irritation during the first week of creatine supplementation. No other participant reported an adverse event. Overall, creatine ingestion during resistance training sessions is a well-tolerated, viable strategy for improving muscle strength and endurance.

The authors report a calculated sample size requirement of 34 participants, but only 26 were recruited. Why the disparity? I think this might have limited results.

Response: Insert Lines 210-214: Based on the sample size calculation, our recruitment goal was 34 participants. However, we were only able to enroll and randomize 26 eligible participants (13 male, 13 female) into the study, which started in August 2019 and ended in December 2019, because the vast majority of participants could only commit to the study during this time frame.

It is strange that the placebo group did not improve performance in several exercises (eg chest press, leg press). The authors state that they have previous shown this training protocol to be effective. I believe the data might be limited by the small sample size in relation to their calculated N (which would have been 17 per group).

Response: Lines 326-328: It is somewhat puzzling that muscle strength and leg press endurance did not increase over 6 weeks of training in the placebo group. However, with only 9 participants, we likely had insufficient power to detect significant changes over time.

Could the lack of an effect in females for chest press strength (lines 250-252) be due to a lack of statistical power? Certainly, a +5 kg increase seems fairly substantial.

Response: Insert Lines 337-339: It is important to note that females on creatine experienced a ~ 5 kg increase in chest press strength over time. Our small sample size may have decreased our ability to detect significant differences with training.

I do think the authors should highlight a little more the potential issue of their sample size in the discussion. They make one brief mention but based upon my comments here, I think it may have substantially compromised some of the findings.

Response: We have highlighted repeatedly in the discussion that our small size likely decreased our ability to detect significant differences between groups.  

Line 208: “28/27 for supplement compliance” – how is this possible? Is this a typo?

Response: The typo has been corrected.

 Conclusions – I suggest also including that future work needs to elucidate the effects of timing on creatine content of muscle.

Response: Insert Lines 363-368: Creatine ingestion during resistance training sessions is a safe and effective strategy to increase muscle strength and endurance in physically active young adults. However, it is unknown whether intra-workout creatine supplementation is more beneficial than consuming creatine at other times of the day during a resistance training program. Future research should determine the effects of the timing of creatine supplementation on muscle mass and performance in a large cohort of physically active young adults.  

Round 2

Reviewer 1 Report

The authors have explained the limitation of the work and revised the manuscript according to the comments.